# Thermodynamic Optimization of the Ethylene Oligomerization Chemical Process

**DOI:** 10.3390/e24050660

**Published:** 2022-05-07

**Authors:** Yajie Yu, Shaojun Xia, Ming Zhao

**Affiliations:** College of Power Engineering, Naval University of Engineering, Wuhan 430033, China; 2019200341@mail.nwpu.edu.cn (Y.Y.); falconyang601@gmail.com (M.Z.)

**Keywords:** finite time thermodynamics, ethylene oligomerization chemical process, specific entropy generation, multi-objective optimization

## Abstract

The use of olefin oligomerization in the synthesis of liquid fuel has broad application prospects in military and civil fields. Here, based on finite time thermodynamics (FTT), an ethylene oligomerization chemical process (EOCP) model with a constant temperature heat source outside the heat exchanger and reactor pipes was established. The process was first optimized with the minimum specific entropy generation rate (SEGR) as the optimization objective, then multi-objective optimization was further performed by utilizing the NSGA-II algorithm with the minimization of the entropy generation rate (EGR) and the maximization of the C_10_H_20_ yield as the optimization objectives. The results showed that the point of the minimum EGR was the same as that of SEGR in the Pareto optimal frontier. The solution obtained using the Shannon entropy decision method had the lowest deviation index, the C_10_H_20_ yield was reduced by 49.46% compared with the point of reference and the EGR and SEGR were reduced by 59.01% and 18.88%, respectively.

## 1. Introduction

Ethylene oligomerization is a reaction that can transform ethylene into nitrogen-free and sulfur-free crude oil, which can be used as clean liquid fuel. It represents a new method for obtaining energy and has good application prospects. However, the popularization and application of this technology are limited by the bottlenecks of high energy consumption and low production rate, which still need further analysis and optimization. The traditional ethylene oligomerization reaction is mainly based on classical thermodynamics and chemical reaction kinetics. Wu et al. [1] calculated the reaction heat, Gibbs free energy and reaction equilibrium constant of an ethylene oligomerization system with a temperature range between 298 K and 700 K, and the influences of temperature and pressure on the equilibrium constant were also studied. Yuan et al. [2] optimized the process conditions of ethylene oligomerization and established the reaction kinetic model. Toch et al. [3,4,5] analyzed the reaction kinetics of ethylene oligomerization and established a one-dimensional heterogeneous industrial reactor model. Chen et al. [6] modeled an olefin oligomerization reactor, which they optimized for the minimum EGR. However, the development of this technology cannot be satisfied by studies only on catalysts and reactors.

In engineering, a chemical process not only includes chemical reactors, but also other components, such as mixers, separators, etc. Thus, the optimization of the whole chemical process is much more reasonable. For a single component, Andresen et al. [7,8] analyzed the minimum EGR of the heat exchange system and studied the mechanical separation process through FTT. Kingston et al. [9] analyzed the EGR of the separation process of air in a low-pressure distillation column. Badescu [10] took ammonia decomposition rate and heat flux as the objective functions and optimized ammonia decomposition reactors by changing the reactor wall temperature, diameter and catalyst particle distribution. Li et al. [11] and Kong et al. [12] established a steam methane reforming reactor and SO_3_ decomposition membrane reactor through FTT and optimized them with the minimum EGR. In terms of using the FTT theory [13,14,15,16] for process optimization, Røsjorde et al. [17] studied the process model for the dehydrogenation of propane and optimized the external heat source temperature of the reactor with the minimum EGR as the optimization objective. Kingston and Razzitte [18,19] established a dimethyl ether synthesis process model—which included a constant temperature reactor, a compressor with different adiabatic efficiencies and a heat exchanger with a constant temperature difference—solved the minimum EGR and the corresponding optimal operating parameters [18], and further studied the coupling of the two reactors [19]. On the basis of Ref. [6], an EOCP model including a mixer, compressor, heat exchanger and reactor will be established using FTT in this paper; the process will be optimized with the minimum SEGR as the first objective of optimization, and a multi-objective optimization will be further performed utilizing the NSGA-II algorithm [20,21,22,23] with minimum EGR and maximum C_10_H_20_ yield as the optimization objectives.

## 2. Physical Model of EOCP

In industry, raw gas is often stored in a compressed state in gas storage tanks, and the compression of the gas from low pressure to high pressure consumes energy. In order to fully consider the energy consumption of the EOCP, in this study, the feed gas was treated with an ambient temperature (298.15 K) and ambient pressure (0.101 MPa) as the initial state.

The EOCP model is divided into four parts, including the mixer, compressor, heat exchanger and reactor, as shown in Figure 1.

According to the reaction equation, the mixed gas used in the process was composed of C_2_H_4_, C_4_H_8_, C_10_H_20_ and N_2_. According to the Peng–Robinson equation [24], the compression factor of the mixed gas was 0.9948, with a deviation of 0.52% from the ideal gas. Therefore, it could be assumed that the mixed gas was an ideal gas. Firstly, the raw reaction materials C_2_H_4_ and C_4_H_8_ and the protective gas N_2_, which does not participate in the reaction, enter the mixer and are combined fully. Secondly, the mixed gas reaches the set temperature and pressure through the action of the heat exchanger and compressor, respectively. Finally, the mixed gas enters the reactor for the chemical reaction. In this process, the working fluid follows the ideal gas equation of state:(1)pV=nRT
where R is the molar gas constant in $J\cdot {\mathrm{mol}}^{-1}\cdot K^{-1}$; p is the pressure in MPa; n is the amount of substance in mol; and T is the temperature of the mixed gas in K. The reaction gas flows through each component and complies with the law of mass conservation:(2)$${\dot{m}}_{\mathrm{in}}={\dot{m}}_{\mathrm{out}}$$

Here, ${\dot{m}}_{\mathrm{in}}$ and ${\dot{m}}_{\mathrm{out}}$ are the inlet and outlet mass flow rates of each component, respectively, in kg/s.

### 2.1. Physical Model of Mixer

It is assumed that the gas mixing process is isothermal and isobaric; thus, the EGR of the mixer can be calculated according to the entropy change rate between the inlet and the outlet, which can be expressed as:(3)$$\Delta {\dot{S}}_{M}=-R\sum_{k}{\dot{n}}_{k}\ln y_{k}$$
where ${\dot{n}}_{k}$ is the molar flow rate of substance *k* in $\mathrm{mol}\cdot s^{-1}$ and $y_{k}$ is the mole fraction of each component substance.

### 2.2. Physical Model of Compressor

It is assumed that the compression process is irreversible adiabatic compression without a chemical reaction. The irreversibility of the compression process can be reflected by the efficiency of the compressor as follows:(4)$$\eta_{C}=\frac{h\left(T_{\mathrm{out}}^{\mathrm{isen}},p_{\mathrm{out}}\right)-h\left(T_{\mathrm{in}},p_{\mathrm{in}}\right)}{h\left(T_{\mathrm{out}},p_{\mathrm{out}}\right)-h\left(T_{\mathrm{in}},p_{\mathrm{in}}\right)}$$
where *h* is the molar enthalpy of the working fluid in J/mol; $p_{\mathrm{in}}$ and $p_{\mathrm{out}}$ are the inlet and outlet pressures of the compressor in MPa, respectively; $T_{\mathrm{out}}^{\mathrm{isen}}$ is the outlet temperature of compressor for the ideal reversible process in K; and $T_{\mathrm{out}}$ is the outlet temperature of compressor for the real irreversible process in K.

The entropy flow rates at the outlet and inlet of the compressor can be calculated according to the following formula:(5)$${\dot{S}}_{C}=F_{T,\mathrm{in}}\sum_{k}y_{k}(S_{k}^{{}^{\circ}}+\int_{T_{0}}^{T}\frac{C_{p,k}}{T}dT)\;-F_{T,\mathrm{in}}n_{k}R\ln\frac{p}{1.01325\times 10^{5}\mathrm{Pa}}$$
where $C_{p,k}$ is the constant pressure molar heat capacity of the substance *k* in $J\cdot {\mathrm{mol}}^{-1}\cdot K^{-1}$. The EGR can be obtained according to the entropy change rate between the inlet and the outlet. The relevant values can be checked according to the chemical manual.

The EGR of the compressor is equal to the entropy change rate of the working fluid between the compressor inlet and outlet:(6)$$\Delta {\dot{S}}_{C}=F_{T,\mathrm{in}}\sum_{k}y_{k}\int_{T_{\mathrm{out}}^{\mathrm{isen}}}^{T_{\mathrm{out}}}\frac{C_{p,k}}{T}dT$$

### 2.3. Physical Model of Heat Exchanger

The heat exchange process is irreversible. The heat exchanger model is shown in Figure 2.

The following assumptions were made for the heat exchanger:

(1) The model of gas in the heat exchange tube is a one-dimensional steady-state model;

(2) There is no chemical reaction in the heat exchange process.

The heat exchange process follows the law of energy conservation, which can be expressed as:(7)$$\frac{dT_{\mathrm{HE}}}{dz}=\frac{\pi d_{\mathrm{HE},i}q_{\mathrm{HE}}}{\sum_{k}F_{k}C_{p,k}}$$
where $q_{\mathrm{HE}}=U_{\mathrm{HE}}(1/T_{\mathrm{HE}}-1/T_{\mathrm{HE},a})$, i.e., the heat transfer process is assumed to follow the linear phenomenological heat transfer law, and $U_{\mathrm{HE}}$ is the heat transfer coefficient in $W\cdot K\cdot m^{-2}$; $T_{\mathrm{HE}}$ and $T_{\mathrm{HE},a}$ are the temperatures of the reactant and the heat source, respectively, in K; $d_{\mathrm{HE},i}$ is the inner diameter of the heat exchanger in m; and $F_{k}$ is the molar flow rate of substance *k* in $\mathrm{mol}\cdot s^{-1}$.

Then, the local EGR can be expressed by the following formula [25,26]:(8)$$\sigma_{\mathrm{HE}}=\pi d_{\mathrm{HE},i}q_{\mathrm{HE}}(\frac{1}{T_{\mathrm{HE}}}-\frac{1}{T_{\mathrm{HE},a}})$$

Finally, the EGR of the heat exchanger can be expressed as:(9)$$\Delta {\dot{S}}_{\mathrm{HE}}=\int_{0}^{L_{\mathrm{HE}}}\sigma_{\mathrm{HE}}dz$$
where $L_{\mathrm{HE}}$ is the length of the heat exchanger in m.

### 2.4. Physical Model of Reactor

A reactor is a piece of equipment that is used to realize the reaction process. In this study, a multi-tube fixed bed reactor was used. The operating conditions of the reaction tubes in each tube bundle are the same; thus, a single tube is taken as an example in this paper. The model of the reactor is shown in Figure 3.

Assuming that the reactor model is in a one-dimensional piston flow steady-state mode [6], there is no radial temperature or concentration gradient, and there is also no axial fluid mixing. There are three parts to EGR in the reactor, which are due to heat transfer, viscous flow and the chemical reaction.

The main reactions Ⅰ and Ⅱ that take place in the reactor:(10)$$C_{2}H_{4}\Leftrightarrow (1/2{)C}_{4}H_{8}\;\Delta_{r}H_{1}<0$$
(11)$$C_{2}H_{4}\Leftrightarrow (1/5{)C}_{10}H_{20}\;\Delta_{r}H_{2}<0$$

As the reaction proceeds, the molar flow rates $F_{k}$ for each reaction component satisfy the following mass conservation equations:(12)$$\frac{dF_{C_{2}H_{4}}}{dz}=-A_{c}\rho_{b}(\eta_{1}r_{1}+\eta_{2}r_{2})$$
(13)$$\frac{dF_{C_{4}H_{8}}}{dz}=\frac{1}{2}A_{c}\rho_{b}\eta_{1}r_{1}$$
(14)$$\frac{dF_{C_{10}H_{20}}}{dz}=\frac{1}{5}A_{c}\rho_{b}\eta_{2}r_{2}$$
(15)$$\frac{dF_{N_{2}}}{dz}=0$$
where $A_{c}$ is the cross-sectional area of the reactor in $m^{2}$; $\rho_{b}$ is the bulk density of the catalytic bed in $\mathrm{kg}\cdot m^{3}$; $\eta_{j}$ is the effective factor of internal diffusion (j=I,II); and $r_{j}$ is the reaction rate (j=I,II) in $\mathrm{mol}\cdot {\mathrm{kg}}^{-1}\cdot s^{-1}$. The reaction rate can be calculated according to the Arrhenius formulas, which can be written as:(16)$$r_{1}=k_{1}(p_{C_{2}H_{4}}-p_{C_{4}H_{8}}^{1/2}/K_{1})$$
(17)$$r_{2}=k_{2}(p_{C_{2}H_{4}}-p_{C_{10}H_{20}}^{1/5}/K_{2})$$
(18)k=(A/1000)exp[−E/(RT)]
where $p_{i}$ is the partial pressure; $k_{i}$ is the reaction rate constant; $A_{j}$ refers to the pre-exponential factor in $\mathrm{mol}\cdot s^{-1}\cdot {\mathrm{kg}}^{-1}\cdot {\mathrm{MPa}}^{-1}$; $K_{i}$ is the reaction equilibrium constant; and $E_{j}$ is the reaction activation energy in $J\cdot {\mathrm{mol}}^{-1}$. The values of $A_{j}$ and $E_{j}$ are given in Table 1, and fit well with the experimental data [6].

The viscous flow process follows the momentum conservation equation [27], which can be expressed as:(19)$$\frac{dp}{dz}=-\left[\frac{150\mu_{m}}{d_{p}^{2}}\frac{{\left(1-\varepsilon \right)}^{2}}{\varepsilon^{3}}+\frac{1.75G}{d_{p}}\frac{1-\varepsilon }{\varepsilon^{3}}\right]v_{m}$$
where ε is the porosity of the catalytic bed; $\mu_{m}$ is the viscosity coefficient of the reactant in $\mathrm{kg}\cdot s^{-1}\cdot m^{-1}$; $d_{p}$ is the diameter of the catalyst particles in m; G is the mass flow rate in $\mathrm{kg}\cdot s^{-1}\cdot m^{-2}$; and $v_{m}$ is the average flow rate of the reactants in m/s.

According to the theory of nonequilibrium thermodynamics, the local EGR of the reactor is [14,15]:(20)$$\sigma_{\mathrm{CR}}=\pi d_{\mathrm{CR},i}U_{\mathrm{CR}}{\left(\frac{1}{T}-\frac{1}{T_{a}}\right)}^{2}+A_{c}v_{m}\left[-\frac{1}{T}\left(\frac{dp}{dz}\right)\right]+A_{c}\rho_{b}\sum_{j}\left[\eta_{j}r_{j}(\frac{\Delta_{r}G_{j}}{T})\right]$$

The EGR of the reactor is the integral of the local EGR along the axial direction:(21)$$\Delta {\dot{S}}_{\mathrm{CR}}=\int_{0}^{L_{\mathrm{CR}}}\sigma_{\mathrm{CR}}dz$$
where $L_{\mathrm{CR}}$ is the length of reactor in m.

## 3. Optimization

The EGR of the EOCP is the sum of the EGR of each component:(22)$$\Delta {\dot{S}}_{\mathrm{tot}}=\Delta {\dot{S}}_{M}+\Delta {\dot{S}}_{C}+\Delta {\dot{S}}_{\mathrm{HE}}+\Delta {\dot{S}}_{\mathrm{CR}}$$

It is desirable for the EGR to be as small as possible while the C_10_H_20_ yield is as large as possible; thus, the specific entropy generation rate can be established as follows [16]:(23)$$\Delta {\dot{S}}_{\mathrm{Spec}}=\frac{\Delta {\dot{S}}_{\mathrm{tot}}}{\Delta F_{C_{10}H_{20}}}$$

In this study, the total molar flow rate at the inlet of the mixer ($F_{T,\mathrm{in}}$) and the inlet pressure of the reactor ($p_{\mathrm{in}}$) were chosen as optimization variables. Firstly, the influence of a single variable change on the EGR was analyzed when the other variables and operation parameters were the same as those of the reference process. Secondly, the process performance was optimized with the help of a genetic algorithm, the optimal variable parameter values for the smallest specific entropy generation rate were calculated, and the calculation results of the multi-objective optimization were compared with the reference process. The value ranges of the optimization variables are given by:(24)$$\begin{array}{l} 2\;\mathrm{MPa}\leq p_{\mathrm{in}}\leq 4.5\;\mathrm{MPa}\\ 0.1\;\mathrm{mol}/s\leq F_{T,\mathrm{in}}\leq 2\;\mathrm{mol}/s \end{array}$$

The mathematical description of multi-objective optimization problem is:(25)$$\left\{ \begin{array}{l} \mathrm{Min}\left(\Delta {\dot{S}}_{\mathrm{tot}}\right)\\ \mathrm{Max}\left(\Delta F_{C_{10}H_{20}}\right) \end{array}\right.$$

The parameters of each component of the reference chemical process are shown in Table 2, Table 3 and Table 4; the parameters listed in the table are taken from Ref. [6].

## 4. Calculation Results and Discussion

### 4.1. Minimization of SEGR

The influence of $p_{\mathrm{in}}$ on $\Delta {\dot{S}}_{\mathrm{tot}}$ and $\Delta F_{C_{10}H_{20}}$ of EOCP is shown in Figure 4. As can be seen from the figure, with an increase in $p_{\mathrm{in}}$, $\Delta F_{C_{10}H_{20}}$ gradually increased, the rate of increase slowed and the maximum value of $\Delta F_{C_{10}H_{20}}$ was obtained near 4.1 MPa before gradually decreasing; $\Delta {\dot{S}}_{\mathrm{tot}}$ increased monotonically. The compressor outlet temperature increased with the increase in $p_{\mathrm{in}}$, and the inlet temperature of the reactor increased under the action of the heat exchanger. The inlet temperature increased with the increase in $p_{\mathrm{in}}$, the reaction rates of exothermic reactions Ⅰ and Ⅱ were decreased, and the yield of C_10_H_20_ was decreased. When $p_{\mathrm{in}}$ was greater than 4.1 MPa, $\Delta F_{C_{10}H_{20}}$ decreased, as the influence of temperature was greater than that of pressure. From 2 MPa to 4.1 MPa, although $\Delta F_{C_{10}H_{20}}$ increased by 12.51%, $\Delta S_{\mathrm{tot}}$ also increased by 16.62%. The influence law of $p_{\mathrm{in}}$ on the values of $\Delta {\dot{S}}_{\mathrm{Spec}}$ and $\Delta F_{C_{10}H_{20}}$ of EOCP is shown in Figure 5. It shows that with an increase in $p_{\mathrm{in}}$, $\Delta {\dot{S}}_{\mathrm{Spec}}$ first decreased and then increased. At about 2.58 MPa, the minimum value of 510.0549 was taken as the minimum point of the EGR per C_10_H_20_ yield. At this time, $\Delta F_{C_{10}H_{20}}$ was 0.0455 mol/s and $\Delta {\dot{S}}_{\mathrm{tot}}$ was 23.2147 W/K.

The influence law of $F_{T,\mathrm{in}}$ on $\Delta {\dot{S}}_{\mathrm{tot}}$ and $\Delta F_{C_{10}H_{20}}$ of EOCP is shown in Figure 6. Both $\Delta {\dot{S}}_{\mathrm{tot}}$ and $\Delta F_{C_{10}H_{20}}$ increased with an increase in $F_{T,\mathrm{in}}$. Among them, $\Delta {\dot{S}}_{\mathrm{tot}}$ increased by 14.36 times and $\Delta F_{C_{10}H_{20}}$ increased by 10.25 times; that is, the rate of increase for $\Delta {\dot{S}}_{\mathrm{tot}}$ was faster. The influence law of $F_{T,\mathrm{in}}$ on $\Delta {\dot{S}}_{\mathrm{Spec}}$ and $\Delta F_{C_{10}H_{20}}$ of EOCP is shown in Figure 7. It shows that $\Delta {\dot{S}}_{\mathrm{Spec}}$ decreased first and then increased with the an increase in $F_{T,\mathrm{in}}$; the minimum value was taken at about 0.119 mol/s. The minimum EGR per C_10_H_20_ yield is the most economical point. At this time, $\Delta F_{C_{10}H_{20}}$ was 0.0081 mol/s and $\Delta {\dot{S}}_{\mathrm{tot}}$ was 3.32623 W/K.

The optimal variable values for the minimum $\Delta {\dot{S}}_{\mathrm{Spec}}$ are shown in Table 5. Compared with the reference process, the optimal $p_{\mathrm{in}}$ increased by 0.9923 MPa and the optimal $F_{T,\mathrm{in}}$ decreased by 0.9 mol/s. It was calculated that $\Delta {\dot{S}}_{\mathrm{Spec}}$ decreased by 27.7%, $\Delta S_{\mathrm{tot}}$ decreased by 89.16% and $\Delta F_{C_{10}H_{20}}$ decreased by 85.01% compared to the reference process. This result indicates that the reduction in the value of $\Delta {\dot{S}}_{\mathrm{Spec}}$ in EOCP is realized by sacrificing part of $\Delta F_{C_{10}H_{20}}$.

### 4.2. Multi-Objective Optimization

Using the PlatEMO toolbox, we set the population number as 100, the optimization objective number as 2, the optimization variable number as 2 and the genetic algebra as 100, and the Pareto optimal frontier of EOCP based on the objectives of $\Delta {\dot{S}}_{\mathrm{tot}}$ minimum and $\Delta F_{C_{10}H_{20}}$ maximum was obtained. LINMAP [28], TOPSIS [29] and Shannon entropy [30] decision-making methods were used, where the weight (w) of $\Delta {\dot{S}}_{\mathrm{tot}}$ and $\Delta F_{C_{10}H_{20}}$ was set to 0.5; the optimization results obtained using different decision-making methods were evaluated with the deviation index from Ref. [31].

For the LINMAP decision-making method, the optimal solution i_opt_ is calculated as:(26)$$PED_{i}=\sqrt{\sum_{i=1}^{100}\left(F_{i,j}-F_{j}^{\mathrm{positive}}\right)}$$
(27)$$i_{\mathrm{opt}}\in \min\left\{PED_{i}^{}\right\}$$
where $F_{j}$ is the j-th target value, positive is the positive ideal point and *PED*_i_ is the Euclidean distance between the i-th feasible solution and the positive ideal point. The LINMAP decision-making method calculates the closest point to the positive ideal point.

For the TOPSIS decision-making method, the optimal solution i_opt_ is calculated as:(28)$$NED_{i}=\sqrt{\sum_{i=1}^{100}\left(F_{i,j}-F_{j}^{\mathrm{negative}}\right)}$$
$$Y_{{}_{i}}=\frac{NED_{i}}{PED_{i}+NED_{i}}$$
(29)$$i_{\mathrm{opt}}\in \max\left\{Y_{i}^{}\right\}$$
where negative is the negative ideal point and *NED*_i_ is the Euclidean distance between the i-th feasible solution and the negative ideal point. The TOPSIS decision-making method determines the point farthest from the negative ideal point.

For the Shannon entropy decision-making method, *F*_i,j_ needs to be normalized first:(30)$$P_{i,j}=\frac{F_{i,j}}{\sum_{i=1}^{100}F_{i,j}}$$

The Shannon entropy of the j-th target can be calculated by the following formula:(31)$$SE_{j}=-\frac{1}{\ln100}\sum_{i=1}^{100}P_{i,j}\ln P_{i,j}$$

The optimal solution i_opt_ is calculated as:(32)$$w_{j}=(1-SE_{j})/\sum_{j=1}^{2}(1-SE_{j})$$
(33)$$i_{\mathrm{opt}}\in \min\left\{P_{i,j}\cdot w_{j}\right\}$$
where *w*_j_ is the weight of the j-th target.

The calculation formula for the deviation index is:(34)$$D=\frac{\sqrt{{\sum_{j=1}^{2}\left(\frac{F_{j}^{\mathrm{opt}}-F_{j}^{\mathrm{positive}}}{\sqrt{\sum_{i=1}^{100}F_{i,j}}}\right)}^{2}}}{\sqrt{{\sum_{j=1}^{2}\left(\frac{F_{j}^{\mathrm{opt}}-F_{j}^{\mathrm{positive}}}{\sqrt{\sum_{i=1}^{100}F_{i,j}}}\right)}^{2}}-\sqrt{{\sum_{j=1}^{2}\left(\frac{F_{j}^{\mathrm{opt}}-F_{j}^{\mathrm{negative}}}{\sqrt{\sum_{i=1}^{100}F_{i,j}}}\right)}^{2}}}$$
where opt denotes the decision point. The deviation index is the ratio of the distance between the decision point and the positive ideal point to the sum of the distance between the decision point and the positive and negative ideal point. The deviation index can be used to evaluate the quality of the decision point. The smaller the deviation index, the closer the decision point is to the positive ideal point, and the better the decision point is.

The Pareto optimal frontier of EOCP is shown in Figure 8. The $\Delta {\dot{S}}_{\mathrm{tot}}$ and $\Delta F_{C_{10}H_{20}}$ corresponding to the point where $\Delta {\dot{S}}_{\mathrm{tot}}$ is at a minimum and $\Delta F_{C_{10}H_{20}}$ is at a maximum are the least ideal on the Pareto front, as can be seen from Figure 8, which verifies that for the EOCP, the minimum $\Delta {\dot{S}}_{\mathrm{tot}}$ and the maximum $\Delta F_{C_{10}H_{20}}$ cannot be met at the same time, and the optimal solutions can only be obtained under the different degrees of importance of these two objectives. Both the $\Delta {\dot{S}}_{\mathrm{tot}}$ minimum point and $\Delta F_{C_{10}H_{20}}$ maximum point are the result of a single-objective optimization of one objective without considering the other objective. The reference point in the figure divides the Pareto front into two segments: A and B. The reference point (shown in the hexagon) was obtained by calculating the yield of C_10_H_20_ and the EGR of the reference process. Compared with the reference point, the multi-objective optimization solution on segment A had lower $\Delta {\dot{S}}_{\mathrm{tot}}$ and $\Delta F_{C_{10}H_{20}}$ values, while the multi-objective optimization solution on segment B had higher $\Delta {\dot{S}}_{\mathrm{tot}}$ and $\Delta F_{C_{10}H_{20}}$ values.

The distribution of the Pareto front within the variation range of $p_{\mathrm{in}}$ and $F_{T,\mathrm{in}}$ is shown in Figure 9 and Figure 10. Figure 9 shows that in the Pareto front, the optimization range of *p*_in_ was 2–4.5 MPa; however, it was mainly distributed in the range of 2–3.5 MPa—that is, the selection of $p_{\mathrm{in}}$ in this range would be beneficial to the optimization of EOCP. $F_{T,\mathrm{in}}$ was evenly distributed in the whole optimization range, which shows that the selection of $F_{T,\mathrm{in}}$ was used to adjust the opposition between decreasing $\Delta {\dot{S}}_{\mathrm{tot}}$ and increasing $\Delta F_{C_{10}H_{20}}$ at the same time.

The results of the single-objective optimization, reference point and multi-objective optimization are shown in Table 6. The deviation indexes corresponding to the results of all the optimizations were smaller than the reference point. In the results of the single-objective optimization, the $\Delta {\dot{S}}_{\mathrm{tot}}$ minimum point and the $\Delta {\dot{S}}_{\mathrm{Spec}}$ minimum point were the same, and the deviation index of the $\Delta F_{C_{10}H_{20}}$ maximum point was the largest in the optimization results, indicating that the optimization effect of $\Delta F_{C_{10}H_{20}}$ was worse than $\Delta {\dot{S}}_{\mathrm{tot}}$. The decrease in $\Delta {\dot{S}}_{\mathrm{Spec}}$ was realized by sacrificing part of $\Delta F_{C_{10}H_{20}}$ to reduce $\Delta {\dot{S}}_{\mathrm{tot}}$. Among the multi-objective optimization results, the values for $\Delta S_{\mathrm{tot}}$ and $\Delta F_{C_{10}H_{20}}$ obtained by LINMAP were the largest, and the corresponding deviation index was also the largest. The values for $\Delta {\dot{S}}_{\mathrm{tot}}$ and $\Delta F_{C_{10}H_{20}}$ obtained with the Shannon entropy decision method were the smallest, and the corresponding deviation index was also the smallest. The values of $\Delta {\dot{S}}_{\mathrm{tot}}$ and $\Delta F_{C_{10}H_{20}}$ and the deviation index determined using the TOPSIS decision method were between the other two decision results. When using the deviation index to evaluate the results, the deviation index of the optimal solution under the Shannon enterprise decision result is the smallest; thus, it is the best optimization scheme. In the results obtained using the Shannon entropy decision method, the value of $p_{\mathrm{in}}$ was 2.0315 MPa and the value of $F_{T,\mathrm{in}}$ was 0.3909 mol/s. Compared with the reference process, the value of $p_{\mathrm{in}}$ was reduced by 32.28% and the value of $F_{T,\mathrm{in}}$ was decreased by 60.91%. The final value of $\Delta {\dot{S}}_{\mathrm{tot}}$ was reduced by 59.01%, and the value of $\Delta F_{C_{10}H_{20}}$ was decreased by 49.46%. The degree of this reaction was lowered by the Shannon entropy decision method, and the EGR was reduced by the decrease in the C_10_H_20_ yield of the process.

## 5. Conclusions

An EOCP model with a constant temperature heat source outside the pipe was established in this paper. The process was optimized with the minimum SEGR. In addition, with the aim of making the EGR is as small as possible and the C_10_H_20_ yield is as large as possible, the multi-objective optimization of EOCP was carried out using the NSGA-Ⅱ algorithm, and the Pareto optimal frontier was obtained. The conclusions can be summarized as follows:
The maximum C_10_H_20_ yield and the minimum EGR cannot be guaranteed. When only $p_{\mathrm{in}}$ was used as the optimization variable, the most economical point was obtained at 4.1 MPa. When only $F_{T,\mathrm{in}}$ was used as the optimization variable, the most economical point was taken at 0.119mol/s.By taking the minimum SEGR as the optimization objective,$\Delta {\dot{S}}_{\mathrm{Spec}}$ was reduced by 27.7%, $\Delta S_{\mathrm{tot}}$ was reduced by 89.16% and $\Delta F_{C_{10}H_{20}}$ was reduced by 85.01% compared to the reference process.A comparison of the decision points indicates that Shannon entropy is the best decision-making method. The corresponding $\Delta {\dot{S}}_{\mathrm{tot}}$ of the decision point was reduced by 59.01%, $\Delta F_{C_{10}H_{20}}$ was reduced by 49.46% and $\Delta S_{\mathrm{Spec}}$ was reduced by 18.88%.


## Figures and Tables

**Figure 1 entropy-24-00660-f001:**
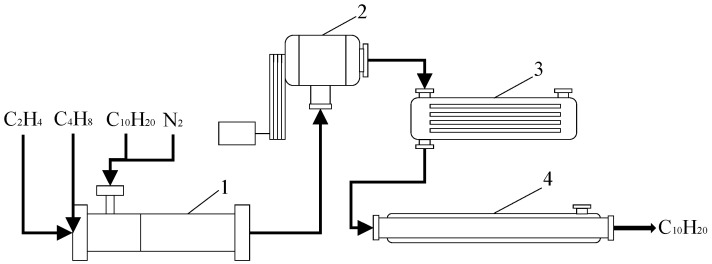
Schematic diagram of EOCP model. 1—mixer; 2—compressor; 3—heat exchanger; 4—reactor.

**Figure 2 entropy-24-00660-f002:**
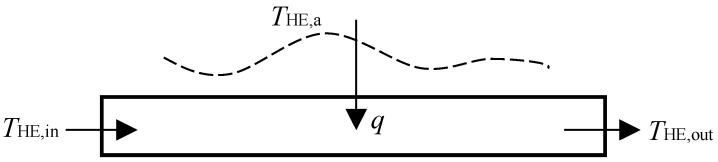
Schematic diagram of heat exchanger model.

**Figure 3 entropy-24-00660-f003:**
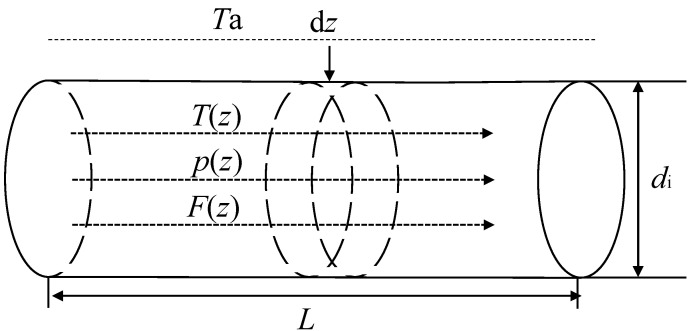
Schematic diagram of one-dimensional plug flow reactor model.

**Figure 4 entropy-24-00660-f004:**
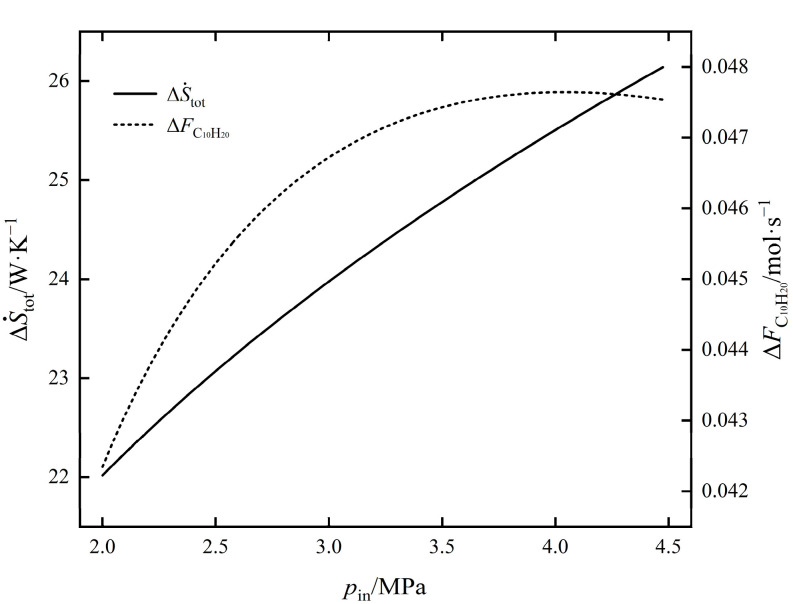
The influence of $p_{\mathrm{in}}$ on $\Delta {\dot{S}}_{\mathrm{tot}}$ and $\Delta F_{C_{10}H_{20}}$.

**Figure 5 entropy-24-00660-f005:**
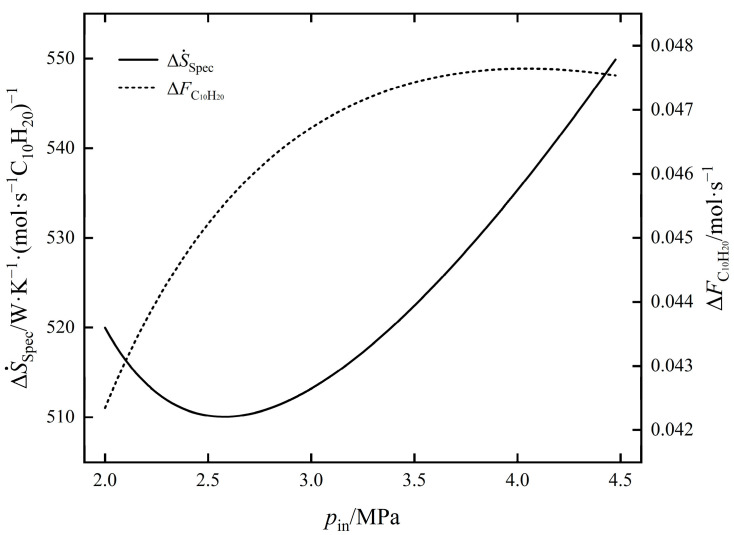
The influence of $p_{\mathrm{in}}$ on $\Delta {\dot{S}}_{\mathrm{Spec}}$ and $\Delta F_{C_{10}H_{20}}$.

**Figure 6 entropy-24-00660-f006:**
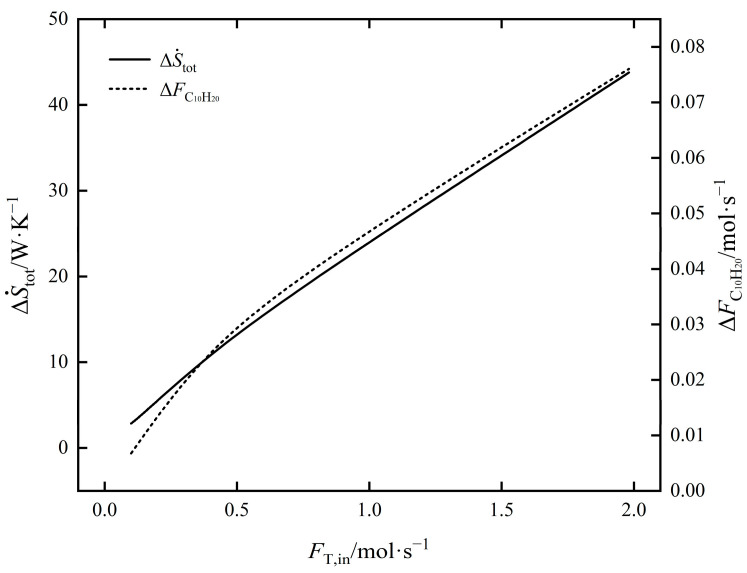
The influence of $F_{T,\mathrm{in}}$ on $\Delta {\dot{S}}_{\mathrm{tot}}$ and $\Delta F_{C_{10}H_{20}}$.

**Figure 7 entropy-24-00660-f007:**
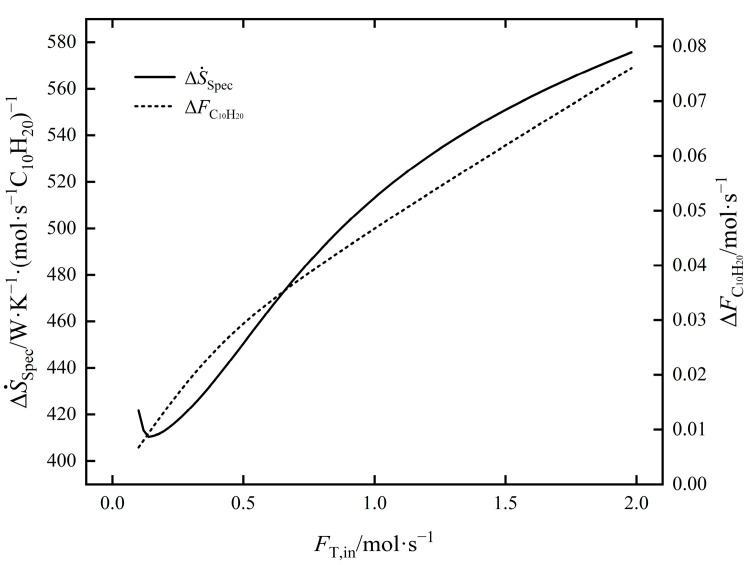
The influence of $F_{T,\mathrm{in}}$ on $\Delta {\dot{S}}_{\mathrm{Spec}}$ and $\Delta F_{C_{10}H_{20}}$.

**Figure 8 entropy-24-00660-f008:**
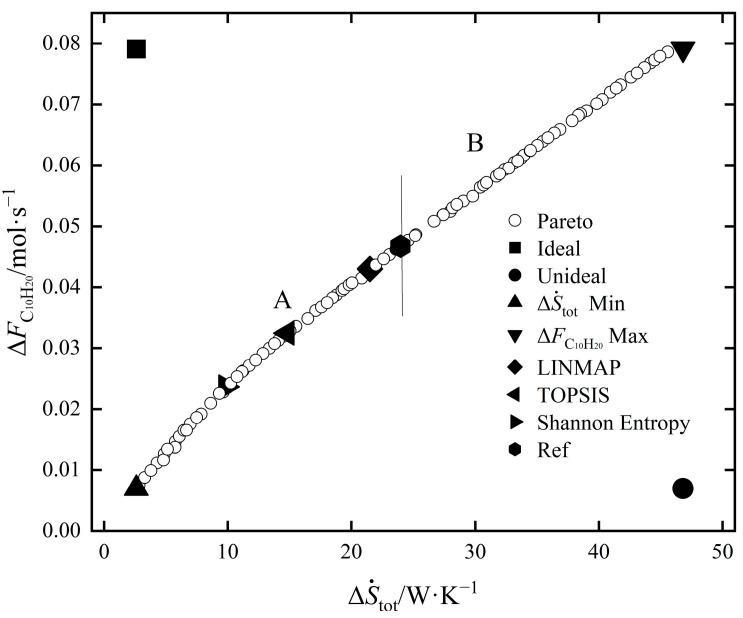
The Pareto front of EOCP.

**Figure 9 entropy-24-00660-f009:**
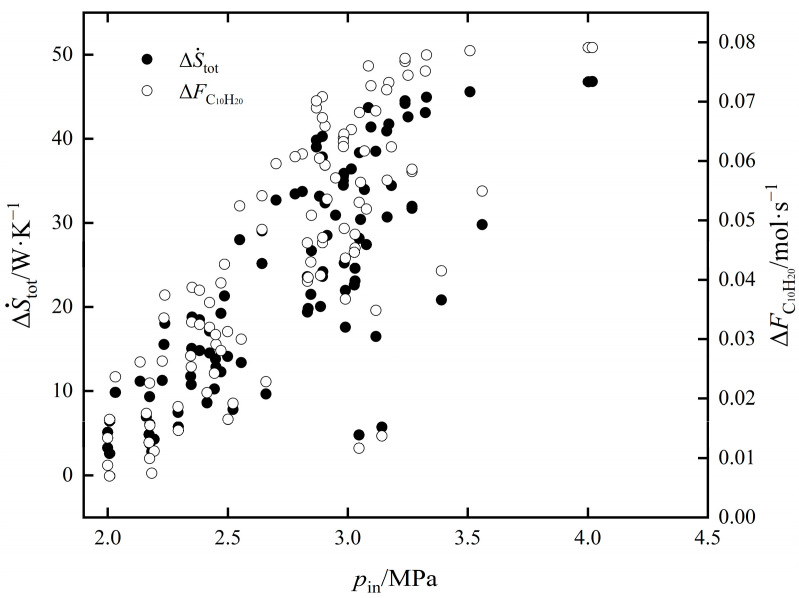
Distribution of the Pareto front within the variation range of $p_{\mathrm{in}}$.

**Figure 10 entropy-24-00660-f010:**
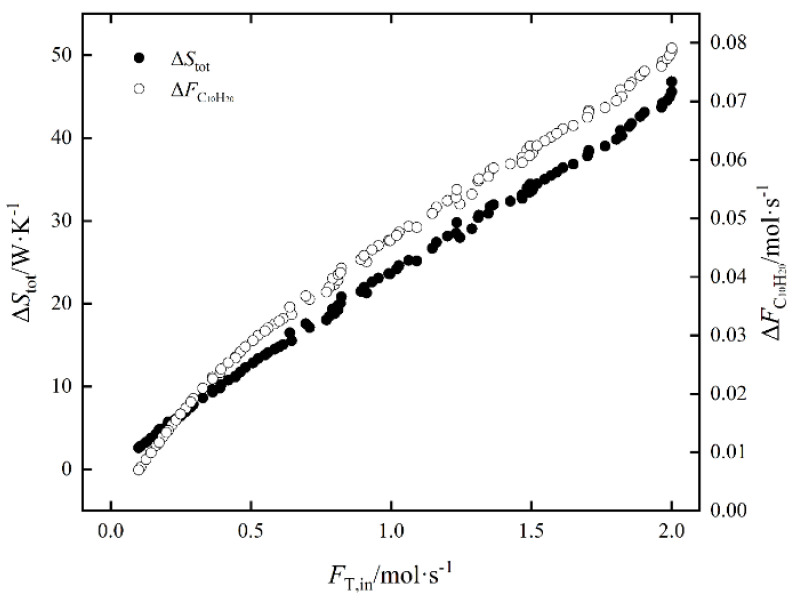
Distribution of the Pareto front within the variation range of $F_{T,\mathrm{in}}$.

**Table 1 entropy-24-00660-t001:** Values of related parameters in the reaction rate equation [6].

Parameters	Reaction 1	Reaction 2
$A_{j}$/mol·(s·kg·MPa)^−1^	19	$4.6\times 10^{3}$
$E_{j}$/J·mol^−1^	56,286	70,116

**Table 2 entropy-24-00660-t002:** Parameters of mixer.

Parameter	Symbol	Value
${\mathrm{Inlet}\;\mathrm{mole}\;\mathrm{fraction}\;\mathrm{of}\;C}_{2}H_{4}$	$y_{C_{2}H_{4},\mathrm{in}}$	0.3
${\mathrm{Inlet}\;\mathrm{mole}\;\mathrm{fraction}\;\mathrm{of}\;C}_{4}H_{8}$	$y_{C_{4}H_{8},\mathrm{in}}$	0.1
${\mathrm{Inlet}\;\mathrm{mole}\;\mathrm{fraction}\;\mathrm{of}\;C}_{10}H_{20}$	$y_{C_{10}H_{20},\mathrm{in}}$	0.1
${\mathrm{Inlet}\;\mathrm{mole}\;\mathrm{fraction}\;\mathrm{of}\;N}_{2}$	$y_{N_{2},\mathrm{in}}$	0.5
Total molar flow rate	$F_{T,\mathrm{in}}$	$1\;\mathrm{mol}\cdot s^{-1}$

**Table 3 entropy-24-00660-t003:** Parameters of heat exchanger.

Parameter	Symbol	Value
Heat transfer coefficient of heat exchanger	$U_{\mathrm{HE}}$	$1.7\times 10^{7}\;W\cdot K\cdot m^{-2}$
Inner diameter of heat exchanger	$d_{\mathrm{HE},i}$	0.08 m
Length of heat exchanger	$L_{\mathrm{HE}}$	5 m

**Table 4 entropy-24-00660-t004:** Parameters of reactor.

Parameter	Symbol	Value
Reactor heat transfer coefficient	$U_{\mathrm{CR}}$	$3\times 10^{7}\;W\cdot K\cdot m^{-2}$
Reactor inner diameter	$d_{\mathrm{CR},i}$	0.08 m
Reactor outer diameter	$d_{\mathrm{CR},o}$	0.084 m
Reactor length	$L_{\mathrm{CR}}$	5 m
Bulk density of catalytic bed	$\rho_{b}$	$800\;\mathrm{kg}\cdot m^{-3}$
External heat source temperature	$T_{a}$	637 K
Catalyst particle diameter	$d_{p}$	0.005 m
Void fraction of catalytic bed	ε	0.45

**Table 5 entropy-24-00660-t005:** Optimization variable values for the minimum SEGR.

Variable	$p_{\mathrm{in}}^{}/\mathrm{MPa}$	$F_{T,\mathrm{in}}^{}/(\mathrm{mol}/s)$
Optimum value	2.0077	0.1

**Table 6 entropy-24-00660-t006:** Comparison of results from the single-objective optimization, reference point and multi-objective optimization.

Optimization Mode	Policy Decision		Optimization Variables	Optimization Objectives		Deviation Index
		$p_{\mathrm{in}}/\mathrm{MPa}$	$F_{T,\mathrm{in}}/\mathrm{mol}\cdot s^{-1}$	$\Delta {\dot{S}}_{\mathrm{tot}}/W\cdot K^{-1}$	$\Delta F_{C_{10}H_{20}}/\mathrm{mol}\cdot s^{-1}$	
Multi-objective	LINMAP	2.8460	0.8922	21.5046	0.0430	0.4616
	TOPSIS	2.3824	0.6012	14.7999	0.0324	0.4542
	Shannon Entropy	2.0315	0.3909	9.8286	0.0236	0.4057
Single objective	$\Delta {\dot{S}}_{\mathrm{tot}}$	2.0077	0.1	2.5982	0.0070	0.420891
	$\Delta {\dot{S}}_{\mathrm{Spec}}$	2.0077	0.1	2.5982	0.0070	0.420891
	$\Delta F_{C_{10}H_{20}}$	4.0184	2	46.8037	0.0791	0.5791
Ref	——	3	1	23.9762	0.0467	0.7521

## Data Availability

The data that support the findings of this study are available from the corresponding author, Y.Y., upon reasonable request.

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
