# Peer review of "Thermodynamic Optimization of the Ethylene Oligomerization Chemical Process"

_entropy, 2022, doi:10.3390/e24050660_

Round 1

Reviewer 1 Report

The authors build on previous models of ethylene oligomerization by considering a more complete model of the process. They explore the model for optimality with regard to entropy production and oligomer production and work out the Pareto frontier between these two objectives. The level of English is very poor and this really gets in the way of communication. Besides this, a number of points are unclear.

How is the reference point chosen? It seems arbitrary. Is it?

What is the Shannon decision? How do any of the three deciding mechanisms choose between Pareto optimal operations?

Where does the Deviation Index come from and why is it a good criterion for deciding?

I recommend sending these questions back to the authors with a plea to paying attention to well formed understandable English. Phrases such as “…are will used …” indicate a level of not caring that is unacceptable. The material appears publish worthy, but it is not easy to tell in the present version.

Author Response

In General

All of the revisions related to the reviewers’ comments were highlighted by red words in the revised manuscript.

The following sentence was added in the “Acknowledgments” of Page 12:

“The authors wish to thank the reviewers for their careful, unbiased, and constructive suggestions, which led to this revised manuscript.”

Reply to the comments of Reviewer 1:

Comment 1: How is the reference point chosen? It seems arbitrary. Is it?

Accepted and Explanation: The reference point is obtained by calculating the reference process, and the parameter values of the reference process are all taken from Ref. [6].

The following sentence was added in the section of “3. Optimization” in Page 6:

“the parameters listed in the table are taken from Ref. [6].”

The following sentence was added in the section of “4.2. Multi objective optimization” in Page 10:

“By calculating the yield of C10H20 and EGR of the reference process, the reference point can be obtained (shown in the hexagon)”

Comment 2: What is the Shannon decision? How do any of the three deciding mechanisms choose between Pareto optimal operations?

Accepted: The following sentences were added in the section of “4.2. Multi objective optimization” in Page 9:

“LINMAP[28], TOPSIS [29] and Shannon Entropy [30] decision-making methods are used in this paper,

For the LINMAP decision-making method, the optimal solution iopt is calculated as:

(23)

(24)

where  is the j-th target values, positive is the positive ideal point. PEDi is the Euclidean distance between the i-th feasible solution and the positive ideal point. LINMAP decision method is to get the closest point to the positive ideal point.

For the TOPSIS decision-making method, the optimal solution iopt is calculated as:

(25)

(26)

where negative is the positive ideal point. NEDi is the Euclidean distance between the i-th feasible solution and the negative ideal point. The TOPSIS decision point is to get the point farthest from the negative ideal point.

For the Shannon Entropy decision-making method, Fi,j needs to be normalized first:

(27)

The Shannon Entropy of the j-th target can be calculated by the following formula:

(28)

The optimal solution iopt is calculated as:

(29)

(30)

where wj is the weight of the j-th target.”

The following references were added in section of “References” in Page 15:

“28.         Sayyaadi H, Mehrabipour R. Efficiency enhancement of a gas turbine cycle using an optimized tubular recuperative heat exchanger. Energy, 2012, 38, 362-375.

  1. Etghani M M, Shojaeefard M H, Khalkhali A, et al. A hybrid method of modified NSGA-II and TOPSIS to optimize performance and emissions of a diesel engine using biodiesel. Appl. Therm. Eng. 2013, 59, 309-315.
  2. Guisado J L, Jimenez-Morales F, Guerra J M. Application of Shannon’s entropy to classify emergent behaviors in a simulation of laser dynamics. Math. Comput. Model, 2005, 42, 847-854. ”

Comment 3: Where does the Deviation Index come from and why is it a good criterion for deciding?

Accepted: Optimization results obtained by different decision-making methods are evaluated by the deviation index from Ref. [31]. The deviation index is the ratio of the distance between the decision point and the positive ideal point to the sum of the distance between the decision point and the positive and negative ideal point. The deviation index can be used to evaluate the quality of the decision point. The smaller the deviation index, the closer the decision point is to the positive ideal point, and the better the decision point is.

The following sentences were added in the section of “4.2. Multi-objective optimization” in Page 9:

…., optimization results obtained by different decision-making methods are evaluated by the deviation index from Ref. [31]”

The following sentences were added in the section of “4.2. Multi-objective optimization” in Page 10:

where opt denotes the decision point. The deviation index is the ratio of the distance between the decision point and the positive ideal point to the sum of the distance between the decision point and the positive and negative ideal point. The deviation index can be used to evaluate the quality of the decision point. The smaller the deviation index, the closer the decision point is to the positive ideal point, and the better the decision point is.

The following reference was added in the section of “References” in Page 15:

“31.        Kumar R, Kaushik S C, Hans R. Multi-objective thermodynamic optimization of an irreversible regenerative Brayton cycle using evolutionary algorithm and decision making. Ain Shams Eng. J., 2016, 7(2): 741-753.”

Reply to the comments of Reviewer 2:

Comment 1 Various acronyms have been used throughout the paper. Perhaps, authors may use another one for ethylene oligomerization process, eg. EOP, since the term appears at many places in the paper.

Accepted: The acronym EOCP was used for the ethylene oligomerization chemical process.

Comment 2: Units of physical quantities have been repeated at various places. These should better be defined in the beginning of the model. Thus, units of common quantities such as pressure, temperature and so on  should not be reiterated to be MPa, K etc

Explanation: In physics, all quantities at all times are a combination of quantity and unit. A mass m may be specified as m = 1.490 kg or as m= 490 g. Those are two different ways of writing the same thing. m = 1.490, i.e. without any unit, makes no sense.

Comment 3: Eq. 3 RHS looks to be rate, while the left hand side is the change. This may be corrected or clarified.

Accepted:  is the entropy generation rate of the mixer, which is equal to the entropy change rate between at the outlet and the inlet, and instructions have been revised for clarity.

The following sentence was added in the section of “2.1. Physical model of mixer” in Page 3:

“….and the EGR of the mixer can be calculated according to the entropy change rate between the inlet and the outlet, i.e.

Comment 4: Eq. 4, quantity 'h' has not been defined, or perhaps missed by the reviewer. It is not there in the Nomenclature.

Accepted: The following sentence was added in the section of “2.1. Physical model of mixer” in Page 3:

“where h is the molar enthalpy of the working fluid in J/mol,”

Comment 5: LHS of Eq. 5 may be Delta S_C.

Accepted: Eq. 5 is the formula for calculating the entropy flow rate of the working fluid at the inlet and outlet of the compressor. The formula for calculating the EGR of the compressor was added.

The following sentence was added in the section of “2.2. Physical model of compressor” in Page 3:

“The EGR of the compressor is equal to the entropy change rate of the working fluid  between the compressor inlet and outlet:

(6)

Comment 6: Before Eqs. 7 and 8, ERG should be replaced with EGR.

Accepted: They were modified in the revised manuscript.

Comment 7: The captions of the figures seem to be very sketchy. Maybe, the caption should be self-contained.

Accepted: The captions of the figures were revised to be self-contained.

The following captions of the figures were revised in the section of “4.2. Multi-objective optimization” in Pages 10 and 11:

“Fig. 8 The Pareto front of EOCP

Fig. 9 The distribution of Pareto front within the variation range of

Fig. 10 The distribution of Pareto front within the variation range of

Table 6 Comparison of results of single-objective optimization and reference point and multi-objective optimization”

The following sentences were added in the section of “4.2. Multi-objective optimization” in Page 10:

“The distribution of Pareto front within the variation range of  and  are showed in Fig. 9 and Fig. 10.

The results of single-objective optimization and reference point and multi-objective optimization are showed in Table 6.”

Reviewer 2 Report

The authors have analyzed a physical model for ethylene oligomerization process, and optimized the performance of this process under minimum entropy generation rate using various algorithms.  The process takes into account the entropy generation during various stages of the process. 

My specific comments to improve the quality of presentation are as follows.

  1. Various acronyms have been used throughout the paper. Perhaps, authors may use another one for ethylene oligomerization process, eg. EOP, since the term appears at many places in the paper.
  2. Units of physical quantities have been repeated at various places. These should better be defined in the beginning of the model. Thus, units of common quantities such as pressure, temperature and so on  should not be reiterated to be MPa, K etc
  3. Eq. 3 RHS looks to be rate, while the left hand side is the change. This may be corrected or clarified.
  4. Eq. 4, quantity 'h' has not been defined, or perhaps missed by the reviewer. It is not there in the Nomenclature.
  5. LHS of Eq. 5 may be Delta S_C.
  6. Before Eqs. 7 and 8, ERG should be replaced with EGR.
  7. The captions of the figures seem to be very sketchy. Maybe, the caption should be self-contained.

Author Response

(The authors gave the same response as above.)

Round 2

Reviewer 2 Report

Authors have considered almost all the comments. However, they have misunderstood comment 2, and  so their reply is irrelevant. In the interest of improving the presentation, i will encourage them to consider comment 2 again. In particular, the question was, why the unit of temperature (and many other common quantities) have been stated again and again in the text. This may be improved.